# Maintaining Preparedness to Severe Though Infrequent Threats—Can It Be Done?

**DOI:** 10.3390/ijerph17072385

**Published:** 2020-03-31

**Authors:** Maya Siman-Tov, Benny Davidson, Bruria Adini

**Affiliations:** 1Department of Emergency Management and Disaster Medicine, School of Public Health, Sackler Faculty of Medicine, Tel Aviv University, Tel Aviv 6139001, Israel; maylev90@gmail.com; 2Division of Emergency & Disaster Management, Ministry of Health, Tel Aviv 6744300, Israel; benny.davidson@MOH.GOV.IL

**Keywords:** emergency preparedness, toxicological/chemical event, evaluation, benchmarks, mass casualty incident

## Abstract

*Background:* A mass casualty incident (MCI) caused by toxicological/chemical materials constitutes a potential though uncommon risk that may cause great devastation. Presentation of casualties exposed to such materials in hospitals, if not immediately identified, may cause secondary contamination resulting in dysfunction of the emergency department. The study examined the impact of a longitudinal evaluation process on the ongoing emergency preparedness of hospitals for toxicological MCIs, over a decade. *Methods:* Emergency preparedness for toxicological incidents of all Israeli hospitals were periodically evaluated, over ten years. The evaluation was based on a structured tool developed to encourage ongoing preparedness of Standard Operating Procedures (SOPs), equipment and infrastructure, knowledge of personnel, and training and exercises. The benchmarks were distributed to all hospitals, to be used as a foundation to build and improve emergency preparedness. Scores were compared within and between hospitals. *Results:* Overall mean scores of emergency preparedness increased over the five measurements from 88 to 95. A significant increase between T1 (first evaluation) and T5 (last evaluation) occurred in SOPs (*p* = 0.006), training and exercises (*p* = 0.003), and in the overall score (*p* = 0.004). No significant changes were found concerning equipment and infrastructure and knowledge; their scores were consistently very high throughout the decade. An interaction effect was found between the cycles of evaluation and the hospitals’ geographical location (F _(1,20)_ = 3.0, *p* = 0.056), proximity to other medical facilities (F _(1,20)_ = 10.0 *p* = 0.005), and type of area (Urban vs. Periphery) (F _(1,20)_ = 13.1, *p* = 0.002). At T5, all hospitals achieved similar high scores of emergency preparedness. *Conclusions:* Use of accessible benchmarks, which clearly delineate what needs to be continually implemented, facilitates an ongoing sustenance of effective levels of emergency preparedness. As this was demonstrated for a risk that does not frequently occur, it may be assumed that it is possible and practical to achieve and maintain emergency preparedness for other potential risks.

## 1. Key Messages

### 1.1. Implications for Policy Makers

As all societies may be afflicted by natural or human-made disasters, it is vital to build a sustainable capacity of emergency preparedness and management for toxicological/hazardous materials events;Available and accessible benchmarks that clearly delineate actions that should be implemented facilitate an ongoing sustenance of emergency preparedness for toxicological/hazardous materials events;Implementing an ongoing evaluation program of emergency preparedness enables us to enhance the capacity to manage both expected or unexpected adversities such as toxicological/hazardous materials events;Such evaluation programs improve general emergency preparedness for toxicological events, and present opportunities to improve varied areas, including equipment and knowledge in all medical centers, and including those located in the periphery;Maintaining a structured evaluation program facilitates resilience and preparedness even to infrequent threats.

### 1.2. Implications for the Public

Populations worldwide are susceptible to varied hazards that may pose a risk to their safety and well-being. We depend on national, regional, and local leadership to implement measures that will protect our societies and ensure that damages are mitigated as much as possible. The present study provides a practical tool that may be beneficial to all administrations as it enables us to build and maintain emergency preparedness for all types of risks, including those that are less familiar and/or common, such as for toxicological/hazardous materials events. By introducing and utilizing an ongoing mechanism of evaluation, any society can enhance its capacity to manage expected or unexpected events. The public should actively demand that its leadership adopt such mechanisms to increase their safety and security.

## 2. Background

The global community is continually exposed to numerous potential emergencies that may evolve from forces of nature or human-induced events. To mitigate potential morbidity and mortality that may result from such emergencies, healthcare systems are required to maintain an ongoing level of preparedness, to ensure resilience and the capacity to effectively respond to any type of mass casualty incident (MCI) that may occur [1]. Many of the hazards occur sporadically or even rarely, but as they may pose a very high risk to both the population and the healthcare system, they cannot be ignored [2].

A significant challenge is how to maintain an effective ongoing preparedness to the varied imminent threats despite their infrequent materialization. Though most mass casualty incidents that occurred globally in the past two decades were of a “conventional” nature, i.e., consisted mostly of trauma victims, numerous events occurred as a result of chemical, toxicological, radiological, or biological (CBRN) threats that were caused either accidentally or were intentionally dispersed among populated communities [3]. The potential devastation that may be caused by the manifestation of CBRN threats necessitates continued readiness, but as they require a varied operational mode compared to routine activities, they present a highly complex challenge to all healthcare systems [4]. Even a single patient that presents to an emergency department (ED) after being exposed to hazardous materials may cause extreme havoc, secondary contamination of medical personnel, full evacuation of the ED, and subsequent interruption or shutdown of operations for a limited or extended period of time [4,5].

In the past decade, the international community witnessed several MCIs that resulted from the use of toxicological and/or chemical agents [5,6]. Although this emergent threat has taxed the capacities of diverse societies and potentially may transpire in any community, to date it is perceived as occurring only infrequently, and thus constitutes a potential though uncommon risk [4].

Since the 11 September 2001 attack in the USA, the interest of governing authorities, as well as of the media and the public, expanded to include chemical and toxicological threats [6]. Many states invest significant resources in building their capacity to manage such events and mitigate their consequences, but nonetheless, it has been stated that such efforts have been too limited and, as of yet, many deficiencies in the level of emergency preparedness are still predominant [7,8,9,10].

### 2.1. Characteristics of Toxicological/Chemical Threats

Emergencies caused by hazardous toxicological/chemical materials may result from “inadvertent residential, industrial, occupational or transportation mishaps; natural disasters; and hazardous-substance releases that are intended to cause harm” [5]. Beyond such risks, the use of hazardous materials may be intentionally directed by humans to cause mass devastation, as part of political or military conflicts or wars, as was displayed in the Syrian civilian war [5,11,12]. Mass casualty incidents that involve hazardous materials may be detrimental to the functional continuity of emergency departments (EDs), leading to chaos, confusion, miscommunication, lack of coordination, and misinformation [13,14]. The presentation of casualties exposed to such materials in the ED, if not immediately identified as a toxicological incident, may cause secondary contamination of both medical personnel and infrastructure, resulting in dysfunction of the ED [4,5]. Exceptional or unpleasant odors or non-specific symptomatology among healthcare workers have also been known to lead to the closing of hospitals’ EDs [7,15,16]. In most cases, especially when there is no significant intoxication of personnel, this is an extreme and unnecessary step. Mass intoxication may result in respiratory failure or respiratory muscles paralysis that can lead to a high prevalence of mortality [6,17]. Nonetheless, many events that occurred in the past did not develop into such severe MCIs; the problem is that a mass psychogenic illness may evolve, enhanced by the confusion and excitement that are often associated with toxicological or chemical events [7,18].

### 2.2. Components of Hospitals’ Preparedness

Collective exposure to toxic materials may lead to a chaotic situation, derived from a lack of communication between varied stakeholders and response teams, scarcity of human and material resources, or deficient infrastructure for provision of patient care [3,19,20]. The severe potential consequences of exposure to toxic materials necessitate the establishment of strategies for life-saving procedures, the implementation of an immediate response, a rapid decontamination process, triage of patients, and on-site and hospital medical treatment procedures [21,22,23]. Protocols and Standard Operating Procedures (SOPs) concerning the treatment of contaminated patients need to be established, the systemic operation of medical entities has to be designed and incorporated in the overall healthcare systems’ disaster plans, and safety procedures have to be defined and integrated in advance to prevent staff’s exposure to toxic materials without adequate protection [7,20].

The need to function under personal protective equipment is challenging and further complicates the provision of life-saving services such as intubation and ventilation of casualties [20,24,25]. However, it is a necessary measure to protect the healthcare providers from secondary contamination prior to the decontamination of the patients exposed to toxicological or chemical agents [26]. The ability to provide airway management, even before decontamination (thus requiring skills to provide these life-saving procedures while dressed in personal protective gear) is challenging and necessitates designated training and exercises [6,27,28,29]. Effective mental health services and risk communication strategies need to be ready and instated to facilitate an immediate management of mass psychic traumas [13,18,30,31].

As an effective response is most often based on routine actions and experiences, building and maintaining an ongoing preparedness to toxicological/chemical risks is not easy to achieve. Emergency responders and authorities should collaborate, collate resources, strengthen networks, and support a unified and coordinated mechanism for responding to such events [5,31,32].

Considering the risk, effective and innovative response models have been developed, and are designed to ensure an appropriate response of medical facilities for such hazards [1,33]. However, the challenge is not only the need to build the preparedness for such events but more so, to maintain a suitable level of readiness over time. It has been presented that severe deficiencies characterize the disaster planning of numerous healthcare systems [1].

### 2.3. Evaluation as a Vital Component to Maintain Readiness

A vital component in building and maintaining preparedness for emergencies is the ongoing evaluation of strengths (that need to be sustained) and areas for improvement (that need to be modified). Towards this goal, varied evaluation tools have been designed, aimed at delineating benchmarks of appropriate actions that need to be implemented to ensure capacity to effectively respond to anticipated and unanticipated emergencies [2,33,34]. However, despite the development and utilization of varied evaluation tools, to date, there is no consensus on which tool is productive in recognizing gaps and areas for improvements and in achieving the overall goal that is the sustenance of a continued level of emergency preparedness [2,33,35].

The Israeli healthcare system has developed and utilized in the past decade an evaluation tool of the readiness for disasters, which is based on measurable, objective parameters that encompass the appraisal of the effectiveness of Standard Operating Procedures (SOPs), training and drills, staff’s knowledge, and infrastructure and equipment [33,34]. The tool formulates the foundation for the healthcare system in promoting an ongoing level of emergency preparedness for varied types of emergencies, including the most common risks (such as mass casualty events) or more infrequent or rare risks (such as toxicological/chemical mass casualty events) [33]. The evaluation tool has been used regularly, through which the overall level of preparedness is assessed in each hospital at a frequency of at least once every two years, and a limited evaluation process (focusing specifically on equipment and infrastructure) is conducted each year [24,33].

The aim of the study was to examine the impact of the longitudinal structured evaluation process on the ongoing level of emergency preparedness of acute-care hospitals for toxicological/chemical mass casualty incidents, over a period of a decade. The preparedness for toxicological/chemical MCIs was chosen as they represent a severe but infrequently occurring risk.

## 3. Method

### 3.1. Study Population

The study population included all 24 acute-care hospitals that operate in Israel that are required by the Ministry of Health (MoH) to be able to admit and treat patients and casualties in MCIs. As part of its regulatory responsibilities and overall authority to ensure the medical system’s preparedness for all potential hazards, the MoH routinely conducts evaluations of emergency preparedness in all acute-care hospitals.

### 3.2. The Evaluation Tool

A designated evaluation tool to measure readiness for varied emergency scenarios, including toxicological/chemical events, was developed by the MoH specifically for the purpose of encouraging and enhancing an ongoing level of emergency preparedness of all acute-care hospitals across the country. The evaluation tool was initially implemented in the healthcare system in 2007 and has since been continually used. It encompasses the major components that were defined and prioritized by an expert panel of over 225 content experts in disaster management, using the Modified Delphi approach [33]. The tool was updated and revised over the years since its conception, following audits of both exercises and actual incidents. The major components of the toxicological/chemical evaluation tool were categorized to four main groups: (1) Standard Operating Procedures (12 benchmarks); (2) equipment and infrastructure (53 benchmarks); (3) knowledge of medical personnel (6 benchmarks); and (4) training and exercises (12 benchmarks). Lessons learned from After Action Reviews that are conducted following mass casualty incidents and/or exercises and drills are also integrated as part of the evaluation process. Each benchmark is measurable, was ranked according to its specific importance or impact on the level of emergency preparedness (high impact, moderate impact, low impact), and was classified as indicators according to four levels of performance (from fully satisfactory to unsatisfactory). These indicators allow us to assess the level of compliance and readiness of the hospital to each defined benchmark. The most vital benchmarks are identified as “red lines,” signifying that if the hospital does not meet the defined standard (its performance is identified as unsatisfactory), it is not approved for operational function, and the Ministry of Health may revoke its accreditation to admit patients. The study was approved by the Ministry of Health.

### 3.3. The Evaluation Process

Since its inception, the evaluation tool has been used by the Israeli Ministry of Health to conduct the periodic assessment of emergency preparedness of all acute-care hospitals in the country. The SOPs, knowledge of staff, and training and exercises are evaluated bi-annually, while the equipment and infrastructure are evaluated annually (to ensure that they are fully operational at any point of time). The SOPs are submitted by each hospital to the MoH for review, while all other components are reviewed and evaluated in a site visit, utilizing approximately 8–10 evaluators that are employees of the MoH’s Emergency Division or the Home Front Command. The evaluators mark the level of performance of each benchmark, as it is observed in the review of the SOPs and/or the site-visit. To date, 5 full cycles of evaluations were made in the healthcare system, utilizing the evaluation tool. Over the years (between cycles), some of the benchmarks were somewhat revised to adapt to changes in policies or national infrastructures, though no significant modifications were noted, and the overall categories remained constant.

The findings of the evaluations that were conducted in the past decade were recorded into a computerized system that calculated the scores of emergency preparedness, considering the relative ranking of the benchmarks in each category. The results were disseminated individually to each respective hospital, delineating the scores that were calculated for each category as well as to the overall emergency preparedness of the facility. The report also included a detailed account of elements that were not found to be satisfactory to encourage their improvement. Systemic findings or deficiencies that were found in several hospitals were distributed to all hospitals (without identifying the specific hospitals in which they were detected), as well as to other stakeholders, to manifest organizational improvements.

### 3.4. Quality Improvement Strategies

The MoH monitors the improvements implemented in the emergency preparedness of the varied hospitals by requiring the medical facilities to submit a report of the steps initiated to promote the needed changes. In case of a “red line” finding, an additional evaluation is scheduled within a period of up to three months from the original assessment process. The “Emergency Management Committee” that is set up in each hospital is responsible for coordinating the implemented quality improvement measures, and for reporting their findings to the hospital’s management as well as to the MoH. The benchmarks that constitute the evaluation tool were distributed to all hospitals several months prior to the first cycle of evaluations, and updates were sent following any revision of the tool so that they may be used by hospital managements and teams as the foundation to build and improve upon their emergency preparedness. Accordingly, the evaluation tool contributes not only to assess the level of emergency preparedness, but actually facilitates an ongoing methodology for quality assurance.

### 3.5. Statistical Analysis

In the past decade, five cycles of evaluations were conducted. The scores of emergency preparedness calculated for each of the acute-care hospitals, following the evaluation process in the five cycles, were used in this study to compare the ongoing levels of readiness for toxicological/chemical events. According to the values adopted in the evaluation process, the scores were defined by the following 4 levels: excellent (≥91%), high (81–90%), medium (65–89%), or low performance (<65%). The levels were determined by the Supreme Health Authority, which is responsible for the preparedness and management of the healthcare system during emergencies. The scores were compared over time, as well as analyzed according to different hospital characteristics such as size, ownership, location, and trauma classification (trauma center I, II, or III as defined by the American College of Surgeons).

Descriptive statistics were used to analyze the characteristics of the sample. Wilcoxon Signed Ranks test were used to compare the differences between T1 (scores of cycle 1) and T5 (scores of cycle 5), including the calculation of the percent change (T5-T1/T1*100) and the effect size (mean differences/standard deviation of the differences). Repeated measure analyses were conducted with an interaction effect to hospital characteristics. All statistical analyses were performed using SPSS software version 25 (SPSS Inc., Chicago, Illinois, USA). *p*-values lower than 0.05 were considered to be statistically significant.

## 4. Results

The study included 23 hospitals that were evaluated at least five times during the past decade (one hospital, in which less than five cycles of evaluations were conducted, was not included in the study). Five hospitals were classified as small (less than 400 beds), eleven medium-sized (400–700 beds), and seven large facilities (>700 beds). Five hospitals were classified as Level I trauma centers (TC), fourteen as Level II TC, and four as Level III TC. Nine of the hospitals are governmental, eight are owned and operated by the largest Health Maintenance Organization, two are municipal, and four are non-governmental (NGO) hospitals. Eight hospitals are located in the southern part of the country, nine in the center, eight in the northern area, and three in the eastern part of the country, in the capital city. Most hospitals are located near other medical facilities (16) and in urban areas (17).

### 4.1. Levels of Emergency Preparedness over the Decade of Evaluations

Figure 1 presents the mean scores of emergency preparedness of the hospitals for a toxicological/chemical event over the decade of evaluations.

The measures present the scores in the four categorized elements (SOPs; equipment and infrastructure; knowledge; and training & exercises) as well as in the overall score of emergency preparedness (calculating all four categories of preparedness). The equipment & infrastructure (95.3 ± 4.2) as well as the knowledge (96.4 ± 8.7) indexes were found to be very high at T1, while the SOP index was presented as high (87.0 ± 18.0), and the training and exercise index was scored lowest (74.2 ± 22.2). An improvement was found in two indexes: SOPs (95.9 ± 5.6) and training & exercises (91.0 ± 9.6), while among equipment and infrastructure (96.1 ± 3.3) and knowledge (97.1 ± 12.8), stability in the high values of the index was found. The overall mean score of emergency preparedness among all the hospitals, over the five periods of measurement, increased from 88 to 95.

The overall score of emergency preparedness for a toxicological/chemical event was higher in T5 compared to T1 in 70% of the hospitals (16 out 23). Among all other hospitals, the overall scores were in the ‘excellent’ category, except for one that was in the ‘high’ category of emergency preparedness. The scores in T1 and T5 are presented in Figure 2.

As presented in Table 1, a significant increase between T1 and T5 occurred in three of the five indexes, including the SOP index (*p* = 0.006), training & exercises index (*p* = 0.003), and the overall index (*p* = 0.004). The most significant change is evident in the training & exercises index, presenting a large effect size of 0.83 and an increase of 23% (from 74 to 91), while the SOP index presented a 10% increase with a medium effect size. No significant changes were found concerning the equipment and infrastructure index and the knowledge index; their scores were found to be very high throughout the decade of evaluation process.

### 4.2. Impact of Hospital Characteristics on Emergency Preparedness Scores

Considering the varied hospitals’ characteristics, an interaction effect was tested between cycles of evaluation (time of measures-T1 vs. T5) and hospital characteristics. An interaction effect was found between hospitals’ geographical location (F _(1,20)_ = 3.0, *p* = 0.056), proximity to medical facility (F _(1,20)_ = 10.0, *p* = 0.005), and type of area (Urban vs. Periphery) (F _(1,20)_ = 13.1, *p* = 0.002).

Consistently, hospitals located at areas in the periphery, secluded from other medical facilities, and hospitals located in the southern region, demonstrated a lower level of overall emergency preparedness scores in T1 compared to hospitals located in urban areas, near other medical facilities, or at the central part of the country. However, at T5, all types of hospitals achieved the same high scores of emergency preparedness.

For example, at T1, hospitals secluded from other medical facilities demonstrated a lower level of overall emergency preparedness scores compared to hospitals located near other medical facilities (79.5 ± 8.2 vs. 92.3 ± 7.1); however, at T5, both types of hospitals, regardless of the proximity to other medical facilities, achieved the same high scores of emergency preparedness (93.3 ± 3.2 vs. 95.8 ± 3.8). See Figure 3.

At T1, hospitals located in the periphery demonstrated a lower level of overall emergency preparedness scores compared to hospitals located in urban areas (78.1 ± 8.1 vs. 92.0 ± 7.0). However, at T5, both types of hospitals, regardless of urban or peripheral location, achieved the same high scores of emergency preparedness (93.4 ± 3.5 vs. 95.6 ± 3.8). See Figure 4.

At T1, hospitals located the central part of the country presented a mean excellent score of emergency preparedness, while the hospitals located in other areas presented lower levels of preparedness. However, at T5, the mean scores of hospitals, regardless of their geographic location, achieved a mean score of over 91% (i.e., presenting excellent levels of emergency preparedness). See Figure 5.

No interaction effects were observed among hospitals characterized with varied levels of trauma centers (I, II, or III), ownership (municipal, governmental, or HMO), or size (small, medium, or large), suggesting that the increase trend was not dependent on such hospital characteristics.

## 5. Discussion

Building and maintaining a continuous appropriate level of preparedness to emergencies has proven to be a challenge to most healthcare systems world-wide [2,24,36,37]. More so, as routine tasks significantly impact the time, logistic, and manpower resources of most systems, developing and sustaining such readiness for events that do not occur frequently, it proves to be even more ambitious and complex. Contrary to the reluctance to invest efforts in ensuring emergency preparedness, the results of not being prepared can be devastating, and can lead to severe costs in human lives, damage to the environment, and collapse of infrastructure, as well as social and political calamities [2,4]. Thus, there is a need to design and implement mechanisms that may encourage both medical facilities and authorities to build their preparedness prior to the materialization of adversities, and to maintain it effectively over time [5,32,38,39]. As there are numerous hazardous materials that may cause release scenarios, there is a need to build the emergency preparedness based on the risk assessment of the severity of hazards, the likelihood of their release, and their potential public health impact.

This study aimed to examine the impact of a longitudinal structured evaluation process on the capacity to build and maintain effective emergency preparedness for severe but infrequent risks over time. It has been previously stated that an ongoing evaluation process facilitates improvement of emergency preparedness for ‘conventional’ MCIs [39], but this has not as yet been sufficiently demonstrated as effective in CBRN events [9,38]. For this purpose, the preparedness of hospitals for a mass casualty incident that involved hazardous materials was chosen and evaluated over a period of a decade. The results clearly demonstrated that the utilization of such an evaluation process is effective in achieving the goal of building and sustaining a high level of competency to manage such events. All hospitals, regardless of their geographic location, proximity to other medical facilities, or their positioning in an urban or periphery area, improved their level of emergency preparedness over time.

A vital component of the quality assurance process that should be emphasized is the assurance that the benchmarks that comprise the evaluation tool are available and accessible to the hospitals at all times. To date, despite the vital need to utilize validated and standard benchmarks for emergency preparedness of medical facilities, no global consensus has been reached concerning a specific tool that delineates indicators that are recommended for use [37,40,41]. Available and accessible evaluation indicators are still lacking in the healthcare emergency preparedness realm [10]. More so, very few tools were examined and found to provide conclusive evidence of effectiveness [42,43]. The main objective of the benchmarks is not simply to provide a score of the level of emergency preparedness (though this is important), but rather to provide a platform for an ongoing process of enhancing and improving readiness to potential emergency situations [10,39,44]. The benchmarks enable administrations to be continuously aware of the expectations of the regulatory authorities, what they need to implement to maintain readiness over time, and what they need to facilitate the ongoing assessment of achievements. Considering the levels of preparedness for toxicological/chemical events, which were found in most hospitals even in the initial cycle of evaluation (especially concerning equipment, infrastructure, and knowledge), it may be assumed that the utilization of the benchmarks facilitated their capacity to build the preparedness and improve it over time.

It is interesting to note that two components of equipment and infrastructure as well as knowledge of staff were found to be highly scored in the baseline evaluation, and these high levels were maintained over time. Equipment and infrastructure for CBRN events are provided to all Israeli hospitals by the MoH. Thus, it is not surprising that the basic level was found to be homogeneously satisfactory among all medical facilities [33,39]. The challenge is keeping them functional (stocked near the emergency department, refreshed continually to avoid their expiration, ensure proper levels of stocks, etc.) over time [45]. As the evaluation concerning these elements was conducted annually, this most probably contributed to their ongoing level of high preparedness.

More surprising is the similar trend of high preparedness that was displayed concerning the knowledge of staff, as most healthcare workers do not often participate in MCIs involving hazardous materials. Building and sustaining a high level of knowledge concerning MCIs that result from toxicological/chemical materials has been found in varied healthcare systems to be complex [5,8]. One possible explanation for this phenomenon is the accessibility of data that characterizes the Israeli acute-care hospitals. Each ED has checklists that delineate required actions upon the emergence of an MCI that involves hazardous materials. The contact information of the National Poison Center is highlighted and information sheets concerning management of each type of material is immediately accessible. The medical teams are, therefore, not required to base their actions on memory or extensive experience, but rather depend on data that are easily accessible to them. One more element that most likely contributes towards the ongoing maintenance of knowledge among the medical teams is the adoption of the ‘all hazards approach’ [33,37]. As managing a CBRN event is based in Israel on the generic plan for ‘conventional’ MCIs, it is simpler for each staff member to know what is expected of them and how they need to function when an event materializes.

It should be emphasized though that knowledge in itself, despite its importance as a basis for function, is not sufficient to ensure performance. This was presented in several studies that presented a lack of correlation between knowledge and performance [46,47]. It is, thus, vital to evaluate beyond the knowledge, and evaluate the actual function of the healthcare workers.

According to the policy of the MoH, the doctrines for each type of emergency scenarios are designed by national advisory committees, comprised of top experts from both the field and regulatory bodies. Each hospital is then required to adapt the national policy to an organizational SOP, according to the facility’s characteristics, resources, infrastructure, etc. Therefore, it can be expected (as was actually displayed) that no significant variance will be identified between the levels of preparedness in the varied hospitals [33,39].

The elements that proved to be most challenging, and hence displayed a lower level of preparedness, were the scores concerning training and exercises. Training and exercises comprise a vital component of most disaster preparedness programs world-wide, aimed at educating, evaluating, and improving policies and plans, as well as performance of personnel [44,48,49]. Concerning this realm, the Israeli hospitals are evaluated on education activities that need to be initiated, conducted, and consistently maintained by the administration and staff themselves in a continuous process [39]. For example, the hospital has to display that designated training programs for MCIs involving hazardous materials are conducted periodically, that over 80% of its staff participated in the training programs in the past two years, that a functional exercise of decontaminating casualties was performed, etc. Contrary to previous findings that expressed doubts concerning the value of drills and exercises [44], the continuous improvement that was evident among all hospitals over the decade of evaluations shows that structured and regular recurrent evaluations contribute towards an ongoing readiness, even for events that do not frequently occur. This finding highlights the benefit of using longitudinal tools that facilitate an ongoing improvement of performance [50,51].

The diversities that were found between the initial scores of the hospitals according to specific characteristics, such as their location in urban or periphery areas, proximity to other medical facilities, geographic setting, etc., are aligned to previous studies that also presented differences concerning plans and policies, availability of resources, training and exercises, and surge capacities [52,53].

The hospitals in the southern district initially presented a lower level of preparedness compared to the hospitals in all other districts. In the last decade, this area has been targeted with sporadic periods of alert, during which the population was exposed to missile attacks. Some of these periods lasted from 3 to 50 days in which a “State of Emergency” was declared. As missile attacks are a risk that significantly varies from hazardous materials, it may explain the lower preparedness that was noted in this region. Possibly, their attention to an additional risk and willingness to invest resources in disaster risk reduction, while investing efforts in combating a more frequently occurring hazard, is more limited [54]. Nonetheless, throughout the evaluation process, all hospitals improved their level of emergency preparedness, regardless of their initial level of readiness, presenting that the existence of benchmarks and “competition” between facilities, even if it is covert, encourages ongoing improvement, even under challenging conditions [39,55].

The important finding is that this diversity is significantly reduced following the longitudinal evaluation process, displaying that it is possible to enhance emergency preparedness of all hospitals, regardless of their respective characteristics.

A significant limitation that should be highlighted is that the evaluation process assesses elements of emergency preparedness rather than an actual management of a toxicological/chemical event. More so, it is not possible to determine from what is presented whether the tool described actually does improve preparedness or whether the results are merely reflecting improved scores.

## 6. Conclusions

MCIs resulting from hazardous materials are a rare occurrence in most acute-care facilities, but when they do materialize, their damage may be significant to both humans and infrastructures. Thus, they realistically represent a scenario that is challenging, from which we can deduce whether it is possible to sustain emergency preparedness over time, despite limited experience of medical staff and managements. The study of a longitudinal evaluation process over a period of a decade presented that the use of available and accessible benchmarks that clearly delineate what needs to be continually implemented facilitates an ongoing sustenance of an effective level of emergency preparedness. As this was demonstrated for a risk that does not frequently occur, it may be assumed that it is possible and practical to achieve and maintain emergency preparedness for other potential risks. It is, thus, recommended that continued evaluation processes be established in healthcare systems encompassing the varied hazards that are identified through risk assessments. Nonetheless, there is a vital need to examine whether similar evaluation processes are productive in different societies, and in relation to diverse emerging risks. A complementary study should be conducted to compare the achievements of the hospitals in the evaluation program to actual performance in real-life events or at the least, in un-notified exercises that will simulate a toxicological/chemical mass casualty incident. More so, there is an additional need to study how citizens, emergency medical technicians (EMTs), and paramedics may also need further training concerning hazardous materials, as the hospitals are only one component of the healthcare system. Even if the hospitals are fully prepared but the EMTs are not adequately trained, they may put the hospitals’ staff at risk.

## Figures and Tables

**Figure 1 ijerph-17-02385-f001:**
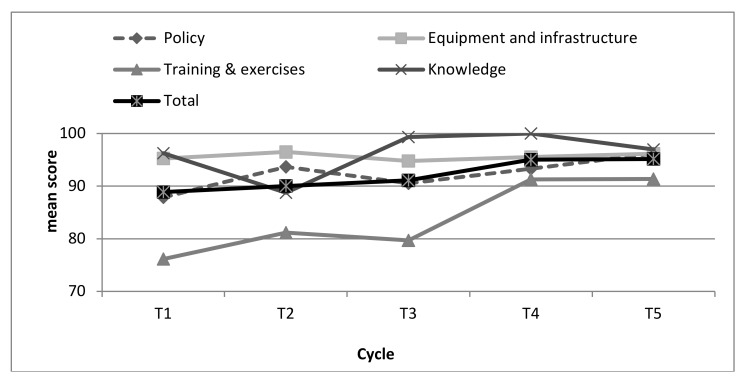
Scores of emergency preparedness for toxicological/chemical events during the five evaluation cycles.

**Figure 2 ijerph-17-02385-f002:**
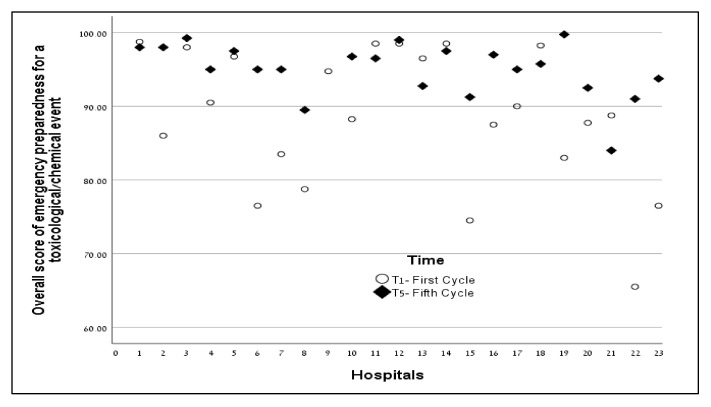
Overall scores of emergency preparedness for toxicological/chemical events in the first and fifth cycle of evaluation.

**Figure 3 ijerph-17-02385-f003:**
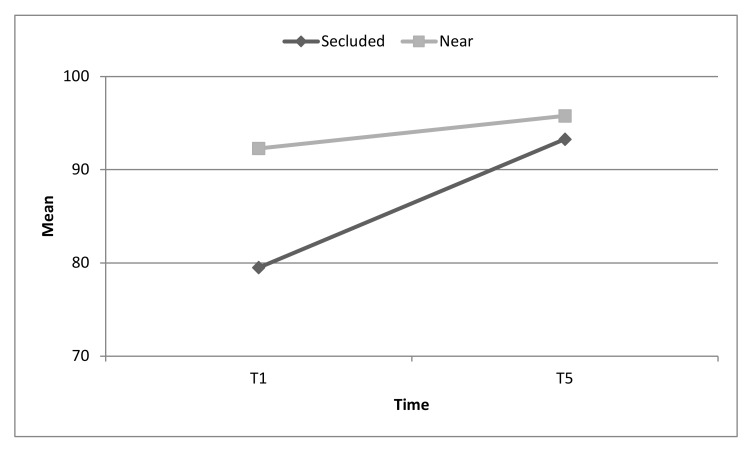
Changes in overall emergency preparedness for toxicological/chemical events by hospitals that are secluded or near other medical facilities. Interaction effect *p* = 0.005.

**Figure 4 ijerph-17-02385-f004:**
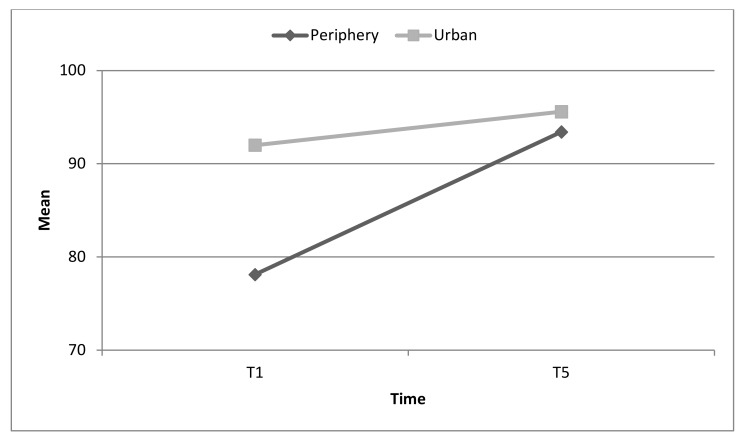
Changes in overall emergency preparedness for toxicological/chemical events by type of area (urban or periphery). Interaction effect *p* = 0.002.

**Figure 5 ijerph-17-02385-f005:**
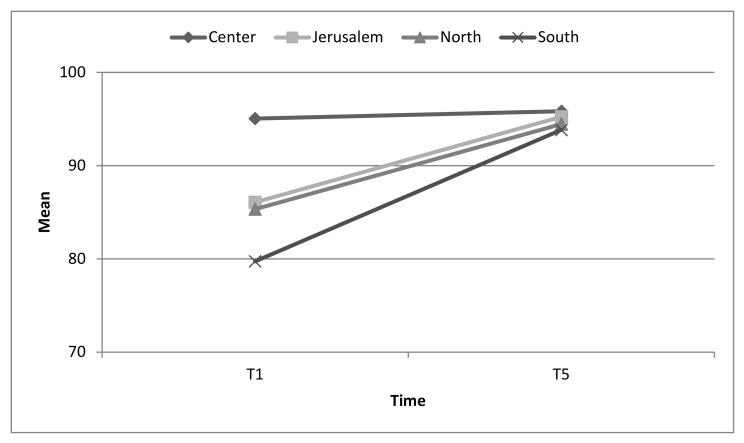
Changes in overall emergency preparedness for toxicological/chemical events measures by geographical location. Interaction effect *p* = 0.056.

**Table 1 ijerph-17-02385-t001:** Differences in emergency preparedness for toxicological/chemical events between T1 (first cycle) and T5 (fifth cycle).

Category	T_1_	T_5_	% Change	Effect Size (Cohen’s d)	*p*-Value
Policy	87.0 ± 18.0 94 (82–98)	95.9 ± 5.6 99.5 (91.8–100)	10.2%	0.48	0.006
Equipment	95.3 ± 4.2 95 (92–99)	96.1 ± 3.3 96 (94–99)	0.8%	0.15	0.464
Drill	74.2 ± 22.2 73 (56–100)	91.0 ± 9.6 94 (83–99)	22.6%	0.83	0.003
Knowledge	96.4 ± 8.7 100 (100–100)	97.1 ± 12.8 100 (100–100)	−0.7%	0.05	0.686
Total	88.2 ± 9.5 89 (83–98)	95.0 ± 3.7 95.4 (93–98)	7.7%	0.80	0.004

Data are presented as mean ± standard deviation or median (Q25–Q75). *p*-value is based on Wilcoxon Signed Ranks Test. Effect size: 0.2—Small, 0.5—Medium, 0.8—Large.

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
