# Peer review of "Maintaining Preparedness to Severe Though Infrequent Threats—Can It Be Done?"

_ijerph, 2020, doi:10.3390/ijerph17072385_

Round 1
Reviewer 1 Report
It's a good scientific work conducted with rigor by the authors.
The data are complete and the conclusions relevant with what is sought.
I have no changes or implementations to ask the authors.
Author Response
We thank the reviewer for reading and approving our article
Reviewer 2 Report
General Comments:
The paper rightly acknowledges that planning and preparedness for CBRN incidents and events is of paramount importance. However, bearing in mind that the number of hazards is large, the number of release scenarios equally as huge, it is not possible to plan and prepare for all eventualities. Accordingly, I would expect that a risk prioritisation approach based on identifying and assessing the severity of hazards, likelihood of release and potential public health impact should underpin any subsequent emergency planning and preparedness activities.
In principle, I would applaud the development of guidelines, standards and benchmarks for ensuring emergency planning and preparedness, as a basis for consistency, governance and excellence. However, it is not possible to determine from what is presented whether the tool described actually does improve preparedness or whether the results are merely reflecting improved scores. This is a key point and the major limitation of this study. It would be useful to the reader see the tool described.
Line 99-Mass Psychogenic Illness may be a more appropriate term than “epidemic hysteria”
Line 132- “Areas for improvement” may be a better term than “weaknesses”
Line 165-Has the tool been updated following audit of actual incidents and events?
Line 183-Seems a rigorous approach to preparedness.
Line 221-On what basis were the levels determined? Are these purely arbitrary?
Line 281-Perhaps some of these scores need to be cross referred to the actual number of incidents during the time of investigation ?
Author Response
The paper rightly acknowledges that planning and preparedness for CBRN incidents and events is of paramount importance. However, bearing in mind that the number of hazards is large, the number of release scenarios equally as huge, it is not possible to plan and prepare for all eventualities. Accordingly, I would expect that a risk prioritisation approach based on identifying and assessing the severity of hazards, likelihood of release and potential public health impact should underpin any subsequent emergency planning and preparedness activities.
Response: The reviewer’s comment was integrated in the manuscript (see after line 320) as follows: As there are numerous hazardous materials which may cause release scenarios, there is a need to build the emergency preparedness based on risk assessment of the severity of hazards, the likelihood of their release and their potential public health impact.
In principle, I would applaud the development of guidelines, standards and benchmarks for ensuring emergency planning and preparedness, as a basis for consistency, governance and excellence. However, it is not possible to determine from what is presented whether the tool described actually does improve preparedness or whether the results are merely reflecting improved scores. This is a key point and the major limitation of this study. It would be useful to the reader see the tool described.
Response: The comment concerning this limitation was added to the discussion section (see after line 413). "It is not possible to relay the full tool as part of the article as it is in Hebrew. Nonetheless, if readers will request this by approaching the authors, the tool will be made available to them should they wish to translate it to their local language"
Line 99-Mass Psychogenic Illness may be a more appropriate term than “epidemic hysteria”
Response: the term was revised to mass psychogenic illness in line 99
Line 132- “Areas for improvement” may be a better term than “weaknesses”
Response: the term was revised to areas for improvement in both line 132 and line 137
Line 165-Has the tool been updated following audit of actual incidents and events?
Response: the tool was updated and revised over the years since its conception, following audits of both exercises and actual incidents. This was added after line 167
Line 183-Seems a rigorous approach to preparedness.
Response: It is a rigorous approach but deemed as necessary to ensure an ongoing preparedness. We added an explanation as to why the equipment and infrastructure are evaluated annually (to ensure that they are fully operational at any point of time). This was added after line 184
Line 221-On what basis were the levels determined? Are these purely arbitrary?
Response: The levels were determined by the Supreme Health Authority which is responsible for the preparedness and management of the healthcare system during emergencies. This was added after line 222
Line 281-Perhaps some of these scores need to be cross referred to the actual number of incidents during the time of investigation?
Response: Israel has had very few mass casualty events that resulted from hazardous/toxicological materials. We cross-referenced those incidents with the scores in the evaluations, and found no correlation between them to the levels of preparedness.
Reviewer 3 Report
This is a great contribution to the literature and the work definitely needs to be published. There is a shortage of information on Mass Casualty Incidents (MCI) that impact hospitals with hazardous materials. Therefore, the topic is definitely of interest to all scholars, policy makers and practitioners.
The authors did a good job describing the threat and explaining why preparedness for exposure to hazardous materials is important for those involved in emergency medicine. This is particularly the case since secondary impacts can shut down a hospital, which may further aggravate a disaster, chemical release or terrorist attack involving HAZMAT.
The development of the research method is to be commended due to the involvement of 225 experts in the Delfi approach. The study is thorough and examined several elements of preparedness in 24 hospitals throughout the nation over a 10 year period.
The findings from the study were clear and revealed that preparedness improved over time, but that there was variation among the different categories of preparedness (e.g., SOPs and training were more successful than equipment and knowledge). The statistics regarding secluded hospitals on the periphery illustrated a potential gap in preparedness.
I commend the authors for acknowledging that there is a need for further consensus on the tool/method. I agree that more case studies are needed as well as continued administration of the existing survey in hospitals over time.
Here are a few recommendations for improvement:
- Change the title by using the word "for" instead of "to."
- The key messages (for both practitioners and the public) could refer specifically to hospital preparedness for MCI patients affected by hazardous materials (rather than just general preparedness). The author may want to note that general preparedness has improved, but then mention opportunities improvement in equipment and knowledge as well as in hospitals in the periphery.
- On line 84 and 85, delete the words "different reasons such as."
- The authors could also indicate in the conclusion that there is an additional need to study how citizens and EMTs/Paramedics may also need further education and training about hazardous materials MCIs. In other words, the hospital is just one part of the system. If the hospital is fully prepared, and the EMTs are not adequately trained they may put doctors and nurses at risk in the Emergency Room. EMTs and Paramedics also need education, training, equipment, communication skills, critical thinking skills, etc.
Overall, this is an excellent study on a topic that needs further elucidation. The authors utilized a solid method to underscore how prepared hospitals are for MCIs involving HAZMAT. With a few small corrections/adjustments, this will be a wonderful contribution to the disaster literature and the practice of emergency medicine.
Author Response
- Change the title by using the word "for" instead of "to."
Response: the title was changed to: Maintaining Preparedness for Severe Though Infrequent Threats – Can it be done?
- The key messages (for both practitioners and the public) could refer specifically to hospital preparedness for MCI patients affected by hazardous materials (rather than just general preparedness).
Response: The specific mention of hazardous materials was integrated in the keywords of both the policy makers and the public
The author may want to note that general preparedness has improved, but then mention opportunities improvement in equipment and knowledge as well as in hospitals in the periphery.
Response: Another key message was included stating - "Such evaluation programs improve general emergency preparedness for toxicological events, and present opportunities to improve varied areas, including equipment and knowledge in all medical centers, including in those located in the periphery".
- On line 84 and 85, delete the words "different reasons such as."
Response: the words “different reasons such as” were deleted
- The authors could also indicate in the conclusion that there is an additional need to study how citizens and EMTs/Paramedics may also need further education and training about hazardous materials MCIs. In other words, the hospital is just one part of the system. If the hospital is fully prepared, and the EMTs are not adequately trained they may put doctors and nurses at risk in the Emergency Room. EMTs and Paramedics also need education, training, equipment, communication skills, critical thinking skills, etc.
Response: The following paragraph was added to the conclusions section, after line 430: “More so, there is an additional need to study how citizens, emergency medical technicians (EMTs) and paramedics may also need further training concerning hazardous materials, as the hospitals are only one component of the healthcare system. Even if the hospitals are fully prepared but the EMTs are not adequately trained, they may put the hospitals’ staff at risk.”